# Significance of the Galectin-8 Immunohistochemical Profile in Ovarian Cancer

**DOI:** 10.3390/biomedicines12020303

**Published:** 2024-01-28

**Authors:** Elena-Roxana Avădănei, Irina-Draga Căruntu, Cornelia Amalinei, Ioana Păvăleanu, Simona-Eliza Giușcă, Andreea Rusu, Ludmila Lozneanu

**Affiliations:** 1Department of Morphofunctional Sciences I—Histology, “Grigore T. Popa” University of Medicine and Pharmacy, 16 University Street, 700115 Iasi, Romania; elena.avadanei@umfiasi.ro (E.-R.A.); irinadragacaruntu@gmail.com (I.-D.C.); simonaelizagiusca@gmail.com (S.-E.G.); andreeadima27@yahoo.com (A.R.); ludmila.lozneanu@umfiasi.ro (L.L.); 2Department of Pathology, Infectious Diseases “Saint Parascheva” Clinical Hospital, 700116 Iasi, Romania; 3Department of Pathology, “Dr. C. I. Parhon” Clinical Hospital, 700503 Iasi, Romania; 4Department of Histopathology, Institute of Legal Medicine, 700455 Iasi, Romania; 5Department of Mother and Child Medicine, “Grigore T. Popa” University of Medicine and Pharmacy, 700115 Iasi, Romania; 6Department of Pathology, “Saint Spiridon” Clinical Emergency County Hospital, 700111 Iasi, Romania

**Keywords:** galectin-8 (Gal-8), ovarian carcinoma (OC), immunohistochemistry (IHC), prognostic factor

## Abstract

Ovarian cancer (OC) still registers a high prevalence in female gynecological pathology. Given the aggressiveness of the tumor and the lack of response to conventional therapies, a current research interest is the identification of new prognostic markers. Gal-8, a member of the galectin family of molecules, involved in tumorigenesis, disease progression, and metastasis, has been assigned as a valuable tumor prognostic factor, and its inhibition may open new perspectives in cancer therapeutic management. Few studies have been carried out so far to evaluate OCs’ galectin profiles. Our study aimed to characterize the Gal-8 profile in different types of ovarian neoplasia and to demonstrate its prognostic value. Our study group comprised 46 cases of OCs that were histologically and immunohistochemically investigated, introduced to Gal-8 immunoreactivity, qualitatively and semi-quantitatively evaluated, and correlated with clinicopathological characteristics. Gal-8 immunoexpression was identified in tumor epithelial cells, showing a dominant nuclear labeling, followed by cytoplasmic and mixed, nuclear, and cytoplasmic labeling. Significant differences between tumor histotypes were found in the statistical analysis between low and high Gal-8 immunoscore levels and clinicopathological features: HGSC (*eng*.= high-grade serous carcinoma) vs. LGSC (*eng.* = low-grade serous carcinoma), pathogenic types (type I vs. type II), and tumor grades. Our results reflect Gal-8 expression variability depending on the histological type and subtype, the progression stages, and the degree of differentiation of ovarian tumors, supporting its value as a prognostic factor. Our findings open perspectives for larger studies to validate our results, along with a potential Gal-8 transformation into a future therapeutic target.

## 1. Introduction

The prevalence of ovarian cancer (OC) is still high in the gynecological pathology spectrum, having the worst prognosis and the highest mortality rate among female malignancies [1,2], standing for the fifth cause of death in women, with a survival rate of less than 50% in 5 years, in the US [3]. 

OC behavior is determined by the tumor’s aggressiveness, frequent impossibility to diagnose it in its curable stage, fast invasion, and dissemination, as well as therapy-resistance, as it is registered in 20% of the patients diagnosed with high-grade serous carcinoma-HGSC [4]. In this context, the identification of new prognosis markers in ovarian tumors is required. The galectin family members stand up among the molecules that may be exploited in this direction. The literature data show that galectin expression is associated with the tumor’s phenotypical heterogeneity and that it may be useful for the molecular sub-classification of aggressive cancers [5,6]. 

Galectin family members (Gal-1, Gal-3, Gal-8, and Gal-9) are considered relevant prognostic factors in OCs. In a study focusing on this particular matter, tumor cells showed Gal-8 expressions, in 8 of 12 cases [4], being detected in both the epithelium and stroma of the ovary and fallopian tube [4,7].

Gal-3, Gal-7, and Gal-8 expressions have been identified in most OC histological subtypes [4,8]. Galectin mostly shows a weak expression in all tumor stages [4,8], while nuclear Gal-8 expression has been correlated with early tumor stages (I–II) [4,8].

The most frequent OC histological subtypes expressing Gal-8 are represented by clear-cell ovarian carcinoma (COC), endometrioid carcinoma (EC), HGSC, and mucinous carcinoma (MOC) [8]. Moreover, following galectin gene expression analyses (*LGALS1*, *LGALS3*, *LGALS4*, *LGALS8*, and *LGALS9*), Gal-8 involvement in ovarian carcinogenesis has been demonstrated, suggesting its value as a new potential therapeutic target [4,7,9].

Additionally, evidence of Gal-8 involvement in the carcinogenesis mechanism has been provided, through cellular migration and proliferation induction and via cellular adhesion and angiogenesis stimulation, added to its involvement in the immune and antitumor inflammatory response modulation [9,10]. Considering the accumulated information, Gal-8 acquires the significance of a tumor prognostic factor, and thus, its inhibition may stand for a new perspective in OC therapy management [11].

By performing a brief literature review, we identified a very small number of studies focused on the prognostic value of Gal-8 in OC. To our knowledge, there are only two previous studies focused on Gal-8’s role in the progression of ovarian cancer [4,12]. Several studies have evaluated different galectins as prognostic markers of survival. For Gal-1, a high expression in the peritumoral stroma is associated with a poor PFS; the cytoplasmic level is correlated with OS, while interstitially expressed Gal-1 is considered an independent prognostic factor for OC. Regarding Gal-3, the published data show that a high Gal-3 expression is correlated with poor FPS, while a reduced nuclear expression and high cytoplasmic expression are correlated with reduced OS [13,14,15].

Gal-7 has been less studied in the context of OC compared to Gal-8. The published data support that a high expression of Gal-7 is associated with a reduction in OS, with Gal-7 being an independent prognostic factor in terms of survival in OC [12].

With regard to Gal-9, the literature data show that a high Gal-9 expression is correlated with a younger age, a lower tumor stage, and a lower grade, and it represents a favorable prognostic factor [12], while epithelial expression and high serum levels are correlated with a lower 5-year DFS and 5-year OS [4].

Given the aforementioned data, our study aimed to characterize the Gal-8 profile in various types of ovarian neoplasms, in association with clinicopathological parameters, such as the patients’ age, the histopathological type, the pathogenic tumor type, as well as the tumor stage and the histological degree of differentiation.

## 2. Materials and Methods

### 2.1. Patients and Tissue Samples

We conducted a retrospective study on 46 patients diagnosed with OC, between 2016–2022, with the Pathology Department, during their hospitalization and therapy at the surgery clinic of “Saint Spiridon” County Clinical Emergency Hospital, Iasi, Romania. The study was approved by the Ethics Committee of the “Grigore T. Popa” University of Medicine and Pharmacy, Iasi (approval no. 12378/June 2015). 

The analysis of the sections of the tumor fragments embedded in paraffin was complied with the legal criteria on the harvesting, handling, and preservation of such specimens.

The study pool included selected OC cases according to their histopathological diagnoses, pathogenic types, tumor stage, degree of differentiation, and patient age. Case classification was performed according to the WHO classification of female genital system tumors [16] and on the dualistic model of pathogenic classification [17,18].

The assessment was carried out by two independent pathologists, with experience in gynecological pathology. Discrepancies were resolved by their agreement.

### 2.2. Immunohistochemical Exam

A paraffin block of each case from our study group was used for immunohistochemistry. The representative paraffin block was selected according to the primary OC morphological examination, identifying the most relevant microscopic characteristics.

Briefly, 4 µm thick tissue sections were sliced from each paraffin block, dewaxed through complete immersion in xylene (3 h at 58 degrees Celsius, followed by 10 min at room temperature) and rehydrated via immersion in four successive alcohol baths of descending concentrations (100%, 90%, 80%, and 70%). The heat-induced antigen retrieval method was applied for antigenic epitope unmasking, using epitope retrieval solution pH 9 (Leica Microsystems GmbH, Wetzlar, Germany), for 30 min. Slides were treated with 3% hydrogen peroxide for endogenous peroxidase activity blocking. The incubation of the primary antibody, anti-Galectin-8 (rabbit monoclonal, ab109519, dilution 1/250, Abcam, Biomedical Campus, Cambridge, UK), was performed overnight, at 4 degrees Celsius. The amplification of the primary antibody reaction was performed using a compatible polymer detection system (ab64261, Abcam, Biomedical Campus, Cambridge, UK), and a visualization of the immunolabeling was obtained using 3.3′-diaminobenzidine tetrahydrochloride chromogen.

Normal ovarian tissue was used as a positive internal control [19], and prostatic adenocarcinoma tissue was used as an external positive control. A negative control was obtained by omitting the primary antibody.

### 2.3. Qualitative and Semi-Quantitative Evaluation of Gal-8

A semi-quantitative assessment of immunohistochemical reactions was performed using a score system previously proposed for breast cancer [20]. The score system took into consideration the sum between the staining intensity and the percentage of Gal-8-positive cells. The immunostaining intensity was considered as 0—negative, 1—weak, 2—moderate, and 3—strong. The percentage of Gal-8-positive cells was classified as 0 for 1–9%, 1 for 10–25%, 2 for 26–50%, 3 for 51–75%, and 4 for ≥76% stained cells. The final scores (the sum between the staining intensity and the percentage of Gal-8-positive cells) between 0–4 were considered low, and the scores between 5–7 were considered high [20].

### 2.4. Statistical Analysis

Statistical data processing was performed using SPSS version 20 (IBM, Armonk, NY, USA) and Microsoft Excel 2016 (Microsoft, Redmond, WA, USA). Continuous variables were expressed as the mean and standard deviation, and the categorical variables were expressed as number (%). The level of association between Gal-8 expression and the clinicopathological characteristics was determined using non-parametric-specific tests (Chi-square test or Fisher’s exact test). The correlation was considered significant when the *p* value was <0.05.

## 3. Results

### 3.1. Clinicopathological Characteristics

The main clinicopathological characteristics of the study group are summarized in Table 1.

The average age of the patients included in the study was 55.56 years. All patients were aged between 37 and 80 years. In total, 25 patients (54.34%) were below the average group age, and 21 patients (45.65%) were above the average group age.

The most frequent histological tumor type was the serous type (26 cases), 18 of them represented by high-grade tumors (39.13%), and only 8 were diagnosed as low-grade cases (17.39%). The serous phenotype was followed by the endometrioid type in nine cases, four of them represented by high-grade tumors (8.69%) and five cases by low-grade tumors (10.86%). Six cases (13.04%) belonged to MOC-type tumors, and five cases (10.86%) belonged to COC-type tumors.

In terms of pathogenic classification, in agreement with the dualistic pathogenic theory, the study pool included 24 cases of OC classified as grade I tumor type (52.17%) and 22 cases (47.82%) classified as grade II tumor type.

Upon consideration of the FIGO/TNM/T stages of ovarian tumors, 22 cases (47.82%) were classified as stage I tumors, only four cases (8.69%) were classified as stage II tumors, and 24 cases (52.17%) were reported as stage III tumors.

In terms of the OC differentiation degree, 17 cases (36.95%) were classified as grade 1 tumor differentiation cases (G1), 12 cases (26.08%) were classified as grade 2 tumor differentiation cases (G2), and 17 cases (36.95%) were classified as grade 3 tumor differentiation cases (G3).

### 3.2. Qualitative Assessment of Gal-8 Immunoexpression

Following Gal-8 expression assessments on all 46 OC cases, a significant variability of the nuclear expression was detected in 37 cases (80%), a cytoplasmic expression in 8 cases (17.39%), and a combined, both nuclear and cytoplasmic expression, in 1 case (2.17%).

Moreover, a significant variability in terms of expression and immunoreaction intensity was noticed in various histopathological types and subtypes of the analyzed OCs as follows: 1 case (2.17%) with intensity 0, 10 cases (21.73%) with intensity 1, 11 cases (23.91%) with intensity 2, and 22 cases (47.82%) with intensity 3 (the maximum intensity level).

The dynamic Gal-8 expression in various OC subtypes is illustrated in Figure 1 and Figure 2. We now underline the high variability of Gal-8 expression, in terms of both pattern and distribution, supporting the influence of tumor differentiation and particular interactions between the tumor cells and the tumor microenvironment in OC development (Figure 1 and Figure 2).

### 3.3. Semi-Quantitative Assessment of Gal-8 Immunoexpression

The Gal-8 immunohistochemistry revealed immunopositivity in all cases and that the percentage of positive cells was higher than 50% in 43 cases (93.47%). The Gal-8 score was high in 33 cases (71.73%), moderate in 12 cases (26.08%), and low in 1 case (2.17%).

The semi-quantitative assessment of Gal-8 immunoreactivity showed the following results: 20 cases (22.98%) with high values, 47 cases (54.02%) with low values, and 20 cases (22.98%) with negative values. Table 1 summarizes the detailed results of the semi-quantitative Gal-8 profile in relation to the main clinicopathological parameters.

The results of the correlation analysis between the high and low score categories of Gal-8 immunoreactivity and the clinicopathological characteristics identified the following features: its expression was predominantly strong (high score) in HGSC in 88.88% of cases; in HGEC, in all cases; in MOC, in 66.66% of cases; in COC, in all cases; in type I, in 54.16% of cases; in type II, in 90.09% of cases; in G2, in 83.33% of cases; and in G3, in 88.23% of cases.

### 3.4. Correlation between Gal-8 Immunoexpression and Clinicopathological Characteristics

We also analyzed the Gal-8 expression in correlation with patient age, histopathological type, pathogenic tumor type (dualistic model), tumor stage, according to the FIGO classification system for OCs and tumor stage (T) of TNM staging, and OC differentiation grade (Table 1).

The analysis of patient age correlated it with tumor cell Gal-8 expression and showed a predominant percentage of patients aged under 55 years, represented by 25 cases, 18 of them (72%) showing a high immunoscore and 7 cases (28%) showing a low immunoscore. Among 21 cases of patients aged over 55 years, 15 cases (71.42%) showed a strong expression, and 6 cases (28.57%) showed a weak expression. Regardless of the age group, a general predominance of the nuclear location of immunoreaction and an obvious majority of cases showing a high score were noticed, without any statistically significant differences between these two age groups (*p* = 0.6369 and *p* = 0.7750, respectively).

From the perspective of the histopathological type, in serous tumors, a statistically significant difference in terms of immunoreaction location was noticed in favor of the nuclear localization (*p* = 0.0360) and of a high Gal-8 immunoexpression score (*p* = 0.0246).

The correlation analysis between the Gal-8 expression score and other OC histotypes, like endometrioid (*p* = 0.1653 and *p* = 0.0845) and MOC and COC (*p* = 0.8529 and *p* = 0.5207), did not reveal any statistically significant associations.

The correlation analysis between the FIGO/TNM/T stage of ovarian tumor and the localization, along with the Gal-8 immunoexpression score, identified several statistically significant differences only in stage I tumors in relation to the predominantly high score of Gal-8 immunoreaction (*p* = 0.0491), or in relation to the nuclear localization of immunoreaction in stage III tumors (*p* = 0.0507). In these latter tumors, an obvious nuclear localization of Gal-8 expression as well as a high immunoreaction scoring were also noticed.

The correlation analysis between Gal-8 expression and tumor grade revealed a statistically significant association between a Gal-8 expression high score and the tumor grade (*p* = 0.0167).

## 4. Discussion

Gal-8 is a molecule that is expressed in both normal body tissues, such as the heart, brain, mammary gland, uterus, placenta, colon, liver, and spleen, and in various types of cancers (breast, ovarian, prostate, bladder, and lung). The normal epithelial tissue and stroma of the ovary and fallopian tubes express the Gal-8 molecule [10,21]. In addition, patients suffering from malignancies show increased titers of galectin molecules in their plasma, serum, and urine, as compared to titers of healthy individuals. Gal-7, Gal-8, and Gal-9 are located in tumor cell cytoplasm and in stroma, while Gal-1, Gal-3, Gal-8, and Gal-9 are located in cells situated in the outskirts of the tumor region [4].

By following the Gal-8 expression in 46 OC cases from our study group, we conducted a qualitative analysis of the Gal-8 immunoreaction pattern in OC and identified a predominantly nuclear location in the tumor epithelial ovarian cells, with a relatively low number of cases where Gal-8 had a strictly cytoplasmic location, and only one case showed some areas that were stained in a mixed manner, both cytoplasmic and nuclear. This particular finding is in agreement with the results obtained by other studies, which achieved the same staining pattern, with the Gal-8 immunolabeling in tumor cells revealing a predominant nuclear expression, followed by cytoplasmic or mitochondrial expression [22], thus underlining the possibility of using Gal-8 molecule exploitation as a screening tool [22]. Moreover, a low Gal-8 cytoplasmic expression corresponds to lymph node dissemination and an advanced tumor stage [10,12,21]. The literature data suggest that an increased cytoplasmic immunoreactivity of Gal-8 is correlated with a favorable prognosis in OC [12].

In terms of the expression and distribution phenotype, our study supports a positive strong intensity in tumor cells, in 71.73% of cases, compared to 26.08% of cases with moderate intensity, and only 2.71% of cases with weak intensity, with a relatively similar profile observed in the literature [12]. Very few studies have been focused on this topic; one of them identified Gal-8 immunoexpression exclusively in neoplastic cells, without any expression in stromal cells [12]. This issue still remains controversial, as it has been shown that patients with tumor disease have increased titers of galectins in plasma and urine compared to healthy individuals [4]. Additionally, Gal-7, Gal-8, and Gal-9 are located both in cytoplasm and plasma, while Gal-1, Gal-3, Gal-8, and Gal-9 are expressed by cells surrounding the tumoral area [4].

The variable results regarding the presence and localization of Gal-8 may be partially due to the technique or the affinity of the different antibodies used [8]. There are no reported specific antibodies targeting a specific Gal-8 isoform until now. Therefore, immunohistochemistry can be used only to observe the total expression of Gal-8-reactive isoforms. Furthermore, it is not clear whether different anti-Gal-8 antibodies have a high binding affinity for a specific Gal-8 isoform [12]. In addition, whether the isoforms have different effects may be determined in future functional studies [8].

A higher incidence of OC in individuals aged under 55 years has been registered, specifically 54.34%, as compared to patients aged above 55 years, specifically 45.65%, without any statistically significant difference between the two age groups considered from the perspective of the Gal-8 immunoreaction score and localization. However, a higher frequency of OCs in people aged above 55 years is reported in the literature data [23]. This difference may be explained by the inclusion in our study of several patients diagnosed with type I tumors, which, according to the pathogenic, dualistic classification, show a high incidence among people under 55 years of age.

A high Gal-8 expression score in patients aged below 55 years was registered in our study, specifically 72%, as compared to patients aged above 55 years, specifically 71.42%, which is a finding supported, from the perspective of histological subtypes, by a predominant serous subtype, followed by endometrioid, MOC, and COC subtypes [10].

According to the WHO classification of female genital system tumors [10] as well as the pathogenic classification [17,18], OCs are classified into different histopathological categories. Ovarian serous carcinoma and endometrioid carcinoma represent the most frequent tumor types included in our pool, 56.52% and 17.39%, respectively, by MOC and COC, representing 13.04% and 10.86%, respectively.

In terms of the assessment of the Gal-8 expression score in different WHO tumor subtypes, our study revealed a high Gal-8 expression in HGSC, in 88.88%; in EC, in 55.55%; in MOC, in 66.66%; and in COC, in 100% of cases. This particular observation is in agreement with the literature data, where Gal-1 expression in stromal cells and Gal-8 and Gal-9 expressions in epithelial cells represent HGSC molecular characteristics [8].

In relation to the histological subtypes, our data comply with the results of other studies, which supported that the most frequent OC types showing positive Gal-8 expression are represented by HGSC, COC, EC, and MOC [10]. Considering other histological subtypes, Gal-8 expression has registered a heterogeneous expression in LGSCs. However, regardless of the tumor subtype, the predominance of a high score has been constantly noticed (71.73%), reflecting Gal-8 involvement in ovarian carcinogenesis, in agreement with the results of another few studies [9].

The dualistic pathogenic model showed the higher prevalence of type I tumors, as compared to type II tumors (54.17% vs. 47.82%), in agreement with the recent literature data [24]. Gal-8 expression registered a high score in type I tumors, at a percentage of 54.16%, as compared to a high score in 90.09% of type II tumors, suggesting that an elevated Gal-8 score may be correlated with a poor prognosis, considering that type II OCs include fallopian tube carcinomas (HGSCs), carcinosarcomas, and undifferentiated carcinomas [17].

The FIGO/TNM system applied in our study showed all the three progression stages of ovarian tumors (I—A, B, C; II—A, B; III—A, B, C), considering that 47.82% were diagnosed as stage I, 8.69% as stage II, and 43.47% as stage III tumors. This progression profile detected in our study shows a significant percentage of cases diagnosed as stage III tumors, complying with the literature data reporting OC diagnoses in very late stages of tumor progression [25]. This pattern of late diagnoses is attributed both to the absence of symptoms and clinical signs in very early tumor stages, as well as to the absence of coherent screening programs [25].

In terms of the tumor stages, the results of the OCs’ Gal-8 immunoreaction are in agreement with other published results [4]. Thus, the Gal-8 molecule shows a stronger expression in stage III tumors (80%), as compared to stage I tumors (59.09%). Accordingly, our study results also support a significant contribution of the Gal-8 molecule in OC progression. This observation is not in agreement with the remarks of another study, which showed that the nuclear expression of Gal-8 is more frequently associated with a lower FIGO stage [12]. In the same study, a Kaplan–Meier analysis showed that patients with high Gal-8 levels had better DFS as well as overall survival [12]. There was no difference in DFS and overall survival regarding the nuclear localization of the Gal-8 immunoreaction. In addition, a multivariate analysis showed that positive Gal-8 expression may be a prognostic factor independent of OC clinical and pathological variables [8,12].

Regarding tumor differentiation grades, the distribution of the Gal-8 immunoexpression was registered in G1 and G3 (each in 36.95%) and G2, in only 26.08%, in our study group, supporting its constant involvement in ovarian carcinogenesis, regardless of the tumor degree of differentiation, either directly or by the possible activation of different proliferative molecular pathways already identified in other tumors [10,11].

The Gal-8 immunoreaction score was high in a large number of G3 tumors (88.23%), followed by G2 tumors (83.33%) and G1 tumors (47.05%). A statistically significant difference of the Gal-8 immunoreaction score in relation to the tumor differentiation grade, between G2 and G3 vs. G1 tumors, was detected (*p* = 0.0167). The results of the Gal-8 expression in correlation with the differentiation grade support, once more, the Gal-8 molecule’s involvement in OC progression, and its valuable prognostic significance.

The Gal-8 expression pattern in the entire study group complies with that observed in the literature data and other international clinical trials focused on the expression of adhesion molecules in different types of tumors [26], with up-regulated or down-regulated expressions, possibly correlated with tumor progression or recurrence, supporting its diagnosis, prognosis, and/or therapeutic relevance in OCs.

Gal-8 expression is correlated with chemoresistance in HGSC [4]. Reported data indicate that Gal-8 expression in HGSC tumor epithelial cells is statistically correlated with a poor 5-year patient survival, chemoresistance, and disease-free interval [4]. Therefore, Gal-8 and Gal-9 can be used for treatment monitoring before, during, and after targeted therapy [4].

Despite the limited number of cases available for our study and, moreover, the large data heterogeneity, in addition to scarce literature information, we consider that our study may open new perspectives in deciphering Gal-8 role’s in the complex mechanisms of ovarian carcinogenesis. In addition, our results provide substantial evidence regarding the significant Gal-8 influence on ovarian tumor behavior, both by its presence in the tumor microenvironment and by its direct action on tumor cells, reflected in the tumor progression modulation.

## 5. Conclusions

Our study results show that the Gal-8 molecule expression varies according to the histological type and subtype, the progression stages, and the differentiation grades of the ovarian tumors, supporting previous literature reports. However, the future extension of the study group cases and the study of the co-expression of other galectins or other adhesion molecules involved in the mechanisms of ovarian carcinogenesis may validate our findings.

According to our findings, the Gal-8 molecule may be considered a prognostic factor, thus opening perspectives for its exploitation as a future therapeutic target in a personalized therapy procedure.

## Figures and Tables

**Figure 1 biomedicines-12-00303-f001:**
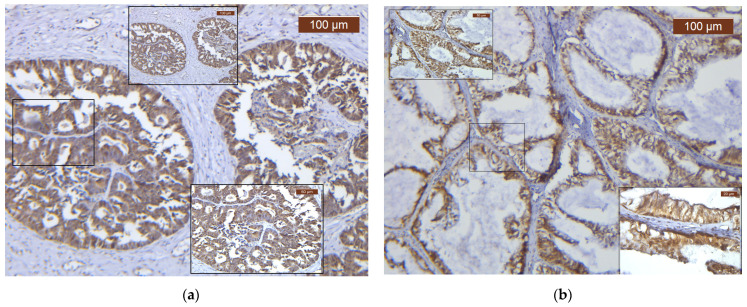
Gal-8 immunostaining in different ovarian carcinoma proliferation patterns. (**a**) Strong diffuse nuclear Gal-8 immunoreaction in HGSC, ×10; (**b**) Moderate diffuse nuclear Gal-8 immunoreaction in MOC, ×10.

**Figure 2 biomedicines-12-00303-f002:**
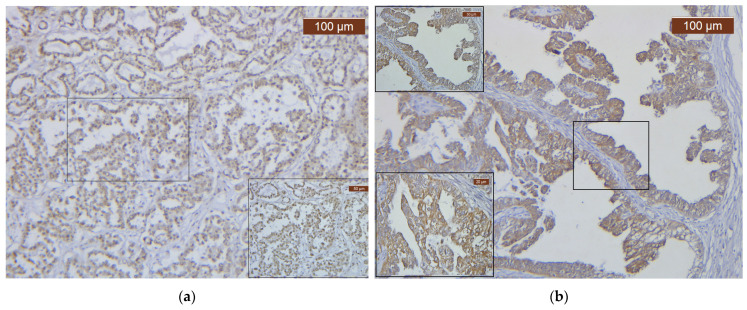
Gal-8 immunostaining in different ovarian carcinoma proliferation patterns. (**a**) Strong diffuse nuclear Gal-8 immunoreaction in COC, ×10; (**b**) Strong diffuse cytoplasmic Gal-8 immunoreaction in LGEC, ×10.

**Table 1 biomedicines-12-00303-t001:** Clinicopathological characteristics and Gal-8 immunoscores in OC patients.

		IHC Reaction Localization				IHC Score			Total
Clinicopathological Characteristics		Nuclear No./%	Cytoplasmic No./%	Mixed No./%	*p* Value	Gal-8 Low Score No./%	Gal-8 High Score No./%	*p* Value *	No./%
**Age**									
≤55 years		20/80	4/16	1/4	0.6369	7/28	18/72	0.7750	25/54.34
>55 years		17/80.95	4/19.04	0		6/28.57	15/71.42		21/45.65
**Histotype**									
HGSC		18/100	0/0	0	0.0360 *	2/11.11	16/88.88	0.0246 *	18/39.13
LGSC		5/62.5	3/37.5	0		5/62.5	3/37.5		8/17.39
HGEC		4/100	0/0	0	0.1653	0/0	4/100	0.0845	4/8.69
LGEC		2/40	2/40	1/20		4/80	1/20		5/10.86
MOC		4/66.66	2/33.33	0	0.8529	2/33.33	4/66.66	0.5207	6/13.04
COC		4/80	1/20	0		0/0	5/100		5/10.86
**Pathogenic** **classification**									
Type I		15/62.5	8/33.33	1/4.16	0.0059 *	11/45.83	13/54.16	0.0148 *	24/52.17
Type II		22/100	0/0	0		2/9.01	20/90.09		22/47.82
**FIGO/TNM(T)**									
I	IA	5/55.55	3/33.33	1/11.11	0.7903	5/55.55	4/44.44	0.0491 *	9/19.56
	IB	3/75	1/25	0		3/75	1/25		4/8.69
	IC	6/66.66	3/33.33	0		1/11.11	8/88.88		9/19.56
II	IIA	3/100	0/0	0	0.6171	0/0	3/100	0.6171	3/6.52
	IIB	1/100	0/0	0		0/0	1/100		1/2.17
	IIC	0/0	0/0	0		0/0	0/0		0/0
III	IIIA	2/67	1/33	0	0.0507 *	2/67	1/33	0.0722	3/6.52
	IIIB	4/100	0/0	0		0/0	4/100		4/8.69
	IIIC	13/100	0/0	0		2/8	11/82		13/28.26
**Tumor grade**									
G1		13/76.47	3/17.64	1/5.88	0.6231	9/52.94	8/47.05	0.0167 *	17/36.95
G2		9/75	3/25	0		2/16.66	10/83.33		12/26.08
G3		15/88.23	2/11.76	0		2/11.76	15/88.23		17/36.95

* Chi-square test (Pearson test)—marked effects are significant at *p* < 0.05. Abbreviations: COC—Clear-Cell Ovarian Carcinoma; FIGO—International Federation of Gynecology and Obstetrics; G1—well-differentiated tumor grade; G2—moderately differentiated tumor grade; G3—poorly differentiated tumor grade; HGEC—High-Grade Endometrioid Carcinoma; HGSC—High-Grade Serous Carcinoma; IHC—Immunohistochemistry; LGEC—Low-Grade Endometrioid Carcinoma; LGSC—Low-Grade Serous Carcinoma; MOC—Mucinous Ovarian Carcinoma.

## Data Availability

The data used to support the findings of this research are available upon request to the authors.

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
