# Peer review of "Significance of the Galectin-8 Immunohistochemical Profile in Ovarian Cancer"

_biomedicines, 2024, doi:10.3390/biomedicines12020303_

Round 1

Reviewer 1 Report

Comments and Suggestions for Authors

The study consists of immunohistochemical analysis of Gal-8 expression in ovarian cancer tissues (paraffin embedded blocks) of different histologic subtypes and different degrees of malignancy (tumor Grade, FIGO stage) The results of statistical analysis are based on the data of Semi-Quantitative Assessment of Gal-8 Immunostaining, which includes the evaluation of staining intensity and the percentage of Gal-8positive cells.

Major comments

1.      First of all, the quality of the presented illustrations does not allow us to evaluate the differences in expression and check their consistency with the conclusions, since the photographs were obviously obtained under different conditions, as evidenced by the very different white balance, etc. In addition, considering that the whole work is based on IHC data, it is necessary to give more examples of staining illustrating different evaluation categories/

2.      The small number of samples in the compared groups also reduces the significance of the findings.

3.    In the context of the literature analysis, it is surprising that there is no reference to the article on a very similar topic, which investigates the expression of Gal-8 in ovarian cancer (also immunohistochemical analysis) "Overall Survival of Ovarian Cancer Patients Is Determined by Expression of Galectins-8 and -9"  doi: 10.3390/ijms19010323. Int J Mol Sci. 2018 Jan; 19(1): 323. 

Conclusion:

Overall, the reviewed manuscript does not meet the level of the journal Biomedicines (Q1 category).

Author Response

Dear reviewer,

Thank you for all your comments.

We have carefully considered the comments and tried our best to address each one of them.

We hope the manuscript meets your standards after careful revision.

Below we provide the point-by-point responses.

  1. Comment 1 First of all, the quality of the presented illustrations does not allow us to evaluate the differences in expression and check their consistency with the conclusions, since the photographs were obviously obtained under different conditions, as evidenced by the very different white balance, etc. In addition, considering that the whole work is based on IHC data, it is necessary to give more examples of staining illustrating different evaluation categories.

Response: We appreciate the reviewer's valuable comments.

Our study was performed according to the protocol described in Section 2. Materials and methods. 2.2. Immunohistochemical study .The figures present in the original manuscript, in section 3. Results, subsection 3.2. Qualitative evaluation of Gal-8 immunoexpression, illustrating firstly the positive localisation of Gal-8 immunoreactivity (exclusively cytoplasmic and exclusively nuclear) and secondly the intensity of immunoreactivity to highlight the presence of immunoreactivity regardless of the morphological type of tumour or despite morphological heterogeneity.

The decision to choose low power field images for illustrative purposes, which made it difficult for the referee to assess Gal-8 immunoreactivity, was motivated at the time of writing the original manuscript by our aim/intention to present (i) aspects of the full morphology spectrum of ovarian tumour lesions, (ii) a range of malignant lesions which, although different in morphology, are all characterized by positive Gal-8 immunoreactivity (Figures 1 and 2).

In particular, in Figure 1 and 2 we have shown a spectrum of Gal-8 positive lesions: 1(a) weak cytoplasmic expression in an HGSC, 1(b) intense nuclear expression in the same tumour type (HGSC), 1(c) moderate diffuse nuclear expression in MOC and 2(a) high intensity Gal-8 immunopositivity diffusely arranged, nuclear in COC, 2(b) and (c) intense and weak expression respectively for Gal-8 in the LGEC subtype.

The reviewer is correct that we did not support our results with a convincing illustration of the variability of Gal-8 immunoexpression intensity. We believe that the difference in the appearance of the images obtained, as pointed out by the reviewer, is due to the fact that the microscopic photographing was not performed in one step, which could have caused the difference observed by the reviewer.

Therefore, in the revised form of the manuscript we excluded one image each from Figure 1 and 2. We kept the rest and, additionally provided images with high power field inserts.

These are the new images and their descriptions:

(a)

(b)

Figure 1. Gal-8 immunostaining in different ovarian carcinoma proliferation patterns. (a) Strong diffuse nuclear Gal-8 immunoreaction in HGSC, x10; (b) Moderate diffuse nuclear Gal-8 immunoreaction in MOC, x10.

(a)

(b)

Figure 2. Gal-8 immunostaining in different ovarian carcinoma proliferation patterns. (a) Strong diffuse nuclear Gal-8 immunoreaction in COC, x10; (b) Strong diffuse cytoplasmic Gal-8 immunoreaction in LGEC, x10;

Either way, it’s possible that we illustrated our results with a reduced number of microscopic images that might not have been too relevant but we assure you that the results and conclusions of our study are supported by the extensive research on the immunereactivity of Gal-8 and setting the expression score on a much larger number of images for each case.

Finally, taking into account your observations on the quality of the illustrations presented, we provide you the original versions of the microscopic images that you will find attached. For a more complete evaluation we express our availability to increase the number of photographs illustrating different evaluation categories.

  1. Comment 2 The small number of samples in the compared groups also reduces the significance of the findings.

Response: We are aware that the small number of samples are generally associated with several disadvantages. So due to the limited amount of affected individuals in our hospital, our goal was to generate hypotheses or gain insights that can be further investigated in larger, more definitive studies.

Also, our target was to identify potential challenges, and refine research protocols before conducting a larger-scale investigation. Even if we have a small number of samples, it may be sufficient to achieve a rich understanding of the Gal-8 expression phenomenon and may be more feasible to provide valuable insights into that group.

Compared to other neoplasia, The International Agency for Research on Cancer’s GLOBOCAN 2020 reported, for Romania, 1909 new cases diagnosed per year, with an indience of 1.9, mortality of 2.1 with 1121 cases/year and 5-year prevalence (all ages) of 5302/100000, we can consider that the overall incidence of ovarian cancer reported in Romania, according to global cancer observatory, is quite low (1.9%) and is associated with a high mortality (2.1%) due to late diagnosis. It should be added that our study protocol excludes patients who received chemotherapy prior to surgery, which significantly reduces the number of included cases.

The fact that our hospital does not have a gynaecological oncological surgery unit also explains the small number of cases selected from patients who addressed to our hospital between 2016 and 2022, added to the pandemic conditions.

Despite the small number of cases, we believe that our results can bring new and valuable information in light of the very small number of studies identified in the literature that focus strictly on Gal-8 expression, moreover they refer to a "cocktail" of galectins and establish prognostic valence by analyzing the relationship between OS and FPS.

  1. Coment 3 In the context of the literature analysis, it is surprising that there is no reference to the article on a very similar topic, which investigates the expression of Gal-8 in ovarian cancer (also immunohistochemical analysis) "Overall Survival of Ovarian Cancer Patients Is Determined by Expression of Galectins-8 and -9" doi: 3390/ijms19010323. Int J Mol Sci. 2018 Jan; 19(1): 323. 

Response: First of all, we consider it an undesirable error to have omitted the cited article. Thank you for your suggestion, especially as it provides us very important data on the Gal-8 expression profile in ovarian cancer. In addition, we believe that it has become a very useful instrument to validate the results of our study. From this point of view, we have seen a lot of similarities in the expression profile of Gal-8, in terms of localization, intensity and the overall level of expression.

In view of the very low number of studies on this topic, this is a very useful tool. The aforementioned article gave us the opportunity to expand the discussion chapter with the main ideas, which can be found in the new version of the article, inserted in the text and highlighted in a different color.

We reformulated the ideas and we look forward to your comments.

Hope we have addressed all the features you asked us to change.

If there are any other changes you consider we should make, please let us know.

Yours sincerely,

Authors

Reviewer 2 Report

Comments and Suggestions for Authors

The article entitled Significance of Galectin-8 Immunohistochemical Profile In Ovarian Cancer serves as a new approach for detecting a novel possible tumor marker in the diagnosis of ovarian cancers. The determination of Galectin-8 expression in different stages of the cancer was assessed by immunohistochemical staining. The study design is scientifically exceptional, however, the data in the manuscript can be considered preliminary results for further detailed examinations because the methods are qualitative and semi-quantitative and the number of included cases is relatively low. 

The presentation of the article is acceptable, the table consistently summarizes the major findings based on different classifications. The figures have good resolution, the images have good quality.

Minor modifications and corrections are suggested in the manuscript:

Row 7: Street should be written with capital.

The reference numbers indicated in the text are written with Arabic and Roman numbers as well. Please, correct the numbering in the text!

Row 51: Gal-Gal-8 should be changed to Gal-8

In the References section, reference No. 15. should be corrected: Please, change the format of the journal name and year. No. 21 has the publication year with normal font also.

Comments on the Quality of English Language

The English language use in the article is good, minor grammatical and typing corrections are suggested.

Author Response

Dear reviewer,

Thank you for all your comments.

There are suggestions for minor changes and corrections in the manuscript and we have made the following corrections:

  1. Regarding first suggestion „Row 7: Street should be written with capital.” – we wrote „Street” with a capital letter

  1. Regarding the number of references indicated in the text, we corrected all of the reference numbers in the text, all are now written with arabic numbers
  2. Regarding „Row 51: Gal-Gal-8 should be changed to Gal-8”, we changed the typo error Gal-Gal-8 to Gal-8
  3. We have taken into account all your valuable suggestions regarding the references section and we have done minor grammatical and typo corrections that were necessary in the English language.

All modifications in the manuscript have been highlighted.

Thank you again for all your advice.

Hope we have addressed all the findings you asked us to change.

If there are any other changes you consider we should make, please let us know.

Yours sincerely,

Authors

Round 2

Reviewer 1 Report

Comments and Suggestions for Authors

Dear authors,

Thank you for your careful work on the manuscript.

The authors have addressed all the comments mentioned in the review and improved the manuscript by enhancing the quality of the figures and interpreting the results more accurately. Additionally,  in accordance with the comments,  the  authors included the data analysis from the previously published paper by H. Schulz et al. in the Introduction and Discussion sections. After these revisions, the manuscript leaves a favorable impression and can be recommended for publication in the journal Biomedicines.